# Low-molecular-weight heparin in the prevention of venous thromboembolism among patients with acute intracerebral hemorrhage: A meta-analysis

**Haizheng Li** [1]*, **Zhiguo Wu**[1], **Hongyu Zhang**[2], **Baohua Qiu**[2], **Yajun Wang**[1]

**1** Department of intervention, Tianjin Medical University Baodi Hospital, Tianjin, China, **2** Department of Cardiovascular Medicine, Tianjin Medical University Baodi Hospital, Tianjin, China

* lihaizhengbdyy@163.com

**Editor:** Sonu Bhaskar, Global Health Neurology Lab / NSW Brain Clot Bank, NSW Health Pathology / Liverpool Hospital and South West Sydney Local Health District / Neurovascular Imaging Lab, Clinical Sciences Stream, Ingham Institute, AUSTRALIA

## Abstract

### Objective

It remains unclear whether low-molecular-weight heparin (LMWH) is effective and safe for intracerebral hemorrhage (ICH) patients. This study presents a meta-analysis for elucidating effect of LMWH on preventing venous thromboembolism (VTE) among ICH patients.

### Methods

Articles were located by systematically searching PubMed, Embase, Web of Science, Cochrane Library, China National Knowledge Infrastructure (CNKI), WANFANG DATA, VIP, and SinoMed databases. The literature was independently screened by two authors, who also extracted data and conducted a qualitative evaluation. With regard to outcomes, their risk ratios (RRs) and 95% confidence intervals (CIs) were computed, and the findings were combined using the random effects model by using Mantel-Haenszel approach.

### Results

30 studies involving 2904 patients were analyzed and compared to control group. According to our findings, early low-dose LMWH, prophylaxis for VTE, was related to the markedly reduced deep vein thrombosis (DVT) (3.6% vs. 17.5%; RR, 0.25; 95% CI, 0.18–0.35; p-value<0.00001) and pulmonary embolism (PE) (0.4% vs. 3.2%; RR, 0.29; 95% CI, 0.14–0.57; p-value = 0.003), while the non-significantly increased hematoma progression (3.8% vs. 3.4%; RR, 1.06; 95% CI, 0.68–1.68; p-value = 0.79) and gastrointestinal bleeding (3.6% vs. 6.1%; RR, 0.63; 95% CI, 0.31–1.28; p-value = 0.20). Also, mortality (14.1% vs. 15.8%; RR, 0.90; 95% CI, 0.63–1.28; p-value = 0.55) did not show any significant difference in LMWH compared with control groups.

**Data Availability Statement:** All relevant data are within the manuscript and its Supporting information files. We confirm our submission contains all raw data required to replicate the results of our study.

**Funding:** The author(s) received no specific funding for this work.

**Competing interests:** The authors have declared that nocompeting interests exist.

## Conclusions

Our meta-analysis suggested that early low-dose of LMWH are safe and effective in ICH patients. More extensive, multicenter, high-quality randomized clinical trials (RCTs) should be conducted to validate the findings and inform clinical practice.

## 1. Introduction

Among patients with intracerebral hemorrhage (ICH), venous thromboembolism (VTE) presents life-threatening consequences and represents a significant global health burden. One study reported that the risks of deep vein thrombosis (DVT) and pulmonary embolism (PE) in ICH patients are 2.4% and 1.1%, respectively [1]. In line with the American Heart Association (AHA) and American Stroke Association (ASA) guidelines, intermittent pneumatic compression (IPC) should be initiated upon admission (class I; level of evidence A), while unfractionated heparin (UFH) or low-molecular-weight heparin (LMWH) can be administered within 1–4 day after admission (class II; level of evidence B) [2]. However, the 2020 guidelines of the Heart and Stroke Foundation of Canada (HSFC) recommend starting LMWH after two days of admission (level of evidence B) [3]. Most of the guidelines that report the use of LMWH and UFH for preventing VTE among ICH patients offer weak recommendations with low-quality evidence [2–5]. However, the American Society of Hematology (ASH) 2018 guidelines provide a strong recommendation for ICH patients, based on moderate certainty of evidence [6]. In view of limited related evidence, only 16.5% of ICH patients receive prophylactic anticoagulation [7]. This is primarily because it is believed that these patients have a high risk of bleeding. Therefore, this work conducted an improved meta-analysis on recent studies (randomized or non-randomized) to elucidate the role of LMWH for VTE prevention in ICH patients. Additionally, the effect of LMWH on DVT, PE, hematoma progression, gastrointestinal bleeding, and mortality was systematically analyzed.

## 2. Methods

### 2.1 Search strategy and screening criteria

The judicious protocol describing specific objectives, search strategy, screening criteria, study quality evaluation, clinical outcomes, and statistical analysis was developed. The protocol was written according to reporting guidelines of Preferred Reporting Items for Systematic Reviews and Meta-analyses (PRISMA). Our protocol was registered in PROSPERO database (registration number: CRD42024525822). PubMed, Embase, Web of Science, Cochrane Library, China National Knowledge Infrastructure (CNKI), WANFANG DATA, VIP, and SinoMed databases were comprehensively searched from inception to November 2023, in addition to a systematic manual search of journal articles. S1 Table shows more details about queries. Further, studies below were included: (1) ICH patients; (2) intervention: LMWH treatment only or LMWH with mechanical treatment [Graduated Compression Stockings (GCS), Intermittent pneumatic compression (IPC)]; (3) comparison: mechanical treatment (GCS, IPC) or not; (4) primary outcomes: DVT, PE, hematoma progression; and secondary outcomes: gastrointestinal bleeding, and mortality; (5) study design: randomized clinical trials (RCTs) and cohort study; (6) publications whose full-texts could be obtained to screen and extract data. In order to remove any irrelevant and non-specific studies, studies below were excluded: (1) Studies involving patients with ICH caused by surgery, traumatic brain injury, or those with

intracranial hemorrhages other than ICH (e.g., subarachnoid hemorrhage, traumatic intracerebral hemorrhage, subdural hematoma, or epidural hematoma); (2) UFH or combined antiplatelet drugs; (3) non-original studies (like review, case report, meta-analysis, or systematic review). Endnote X9 was used to exclude duplicates and screen the literature. Two authors independently selected and assessed the eligibility of English studies or those published in non-English language (namely, Chinese), and any discrepancy was resolved by a third author.

## 2.2 Data collection

Data pertaining to the study design, type of intracerebral hemorrhage, treatment option, dosage, time of onset of treatment, duration of treatment, methods of diagnosis of DVT/PE/hematoma progression, follow-up period, event number, and study participant number were obtained by two investigators. Any disagreement in the data extraction was resolved by adjudicating with the third investigator.

## 2.3 Quality evaluation

Qualitative evaluation of RCTs and non-randomized trials (non-RCTs) was conducted by two investigators using the revised Cochrane risk-of-bias tool (RoB 2) and the Newcastle-Ottawa Scale (high quality 7≤to≤9, moderate quality 4≤to≤6, and low quality <4), respectively. Any conflict in the quality assessment was interceded by a third author.

## 2.4 Endpoints

The primary study endpoints were asymptomatic and symptomatic DVT (diagnosed by clinical symptoms, Doppler ultrasound, venography, and magnetic resonance imaging), PE (diagnosed by clinical symptoms and computed tomography pulmonary angiography), and hematoma progression (diagnosed by clinical symptoms and computed tomography, defined as a ≥33% increase in hematoma volume). The secondary study endpoints were gastrointestinal bleeding and mortality.

## 2.5 Statistical analysis

Review Manager V.5.3 software (Cochrane Collaboration, London, UK) and stata (StataCorp. 2017. Stata Statistical Software: Release 14; StataCorp LLC, College Station, Texas, USA) were employed for data analysis. Continuous variables were described as means (standard deviations), while categorical variables were described as numbers (percentages). We used Mantel-Haenszel approach-based random-effects model for combining results. Effect size was determined by calculating risk ratio (RR) and 95% confidence intervals (CI). To evaluate heterogeneities among the studies, the Mantel-Haenszel method was employed, with p-value = 0.10 indicating statistical significance. Furthermore, Higgins' $I^2$ statistic was employed to compare the extent of heterogeneity (low heterogeneity ≤30%, moderate heterogeneity 30%< to ≤50%, high heterogeneity >50%). For primary endpoints, at least ten studies were conducted for DVT, PE, and hematoma progression. Small-study effects were detected by assessing Egger's test and funnel plot. The trim-and-fill approach was applied to provide an estimate of the treatment effect adjusted for selection bias. Additionally, subgroup analyses based on study design, ICH type, and ICH treatment type (operation or not) were performed. Lastly, the differences among the subgroups were examined in a random-effects model.

# 3. Results

## 3.1 Study screening

Fig 1 exhibits our study screening procedure. There were altogether 13451 records obtained from databases search of citations; in addition, three more articles were found in additional sources. When duplicates were carefully eliminated, 10954 records were acquired and subsequently evaluated. After that, these articles were examined by title- and abstract-reading, with only 52 articles being retained for further study. The full-text of these selected 52 articles was assessed, among which, 30 were finally enrolled for the meta-analysis, while the remaining 22 were excluded. Table 1 summarizes the designs of these 30 qualified articles. The remaining 22

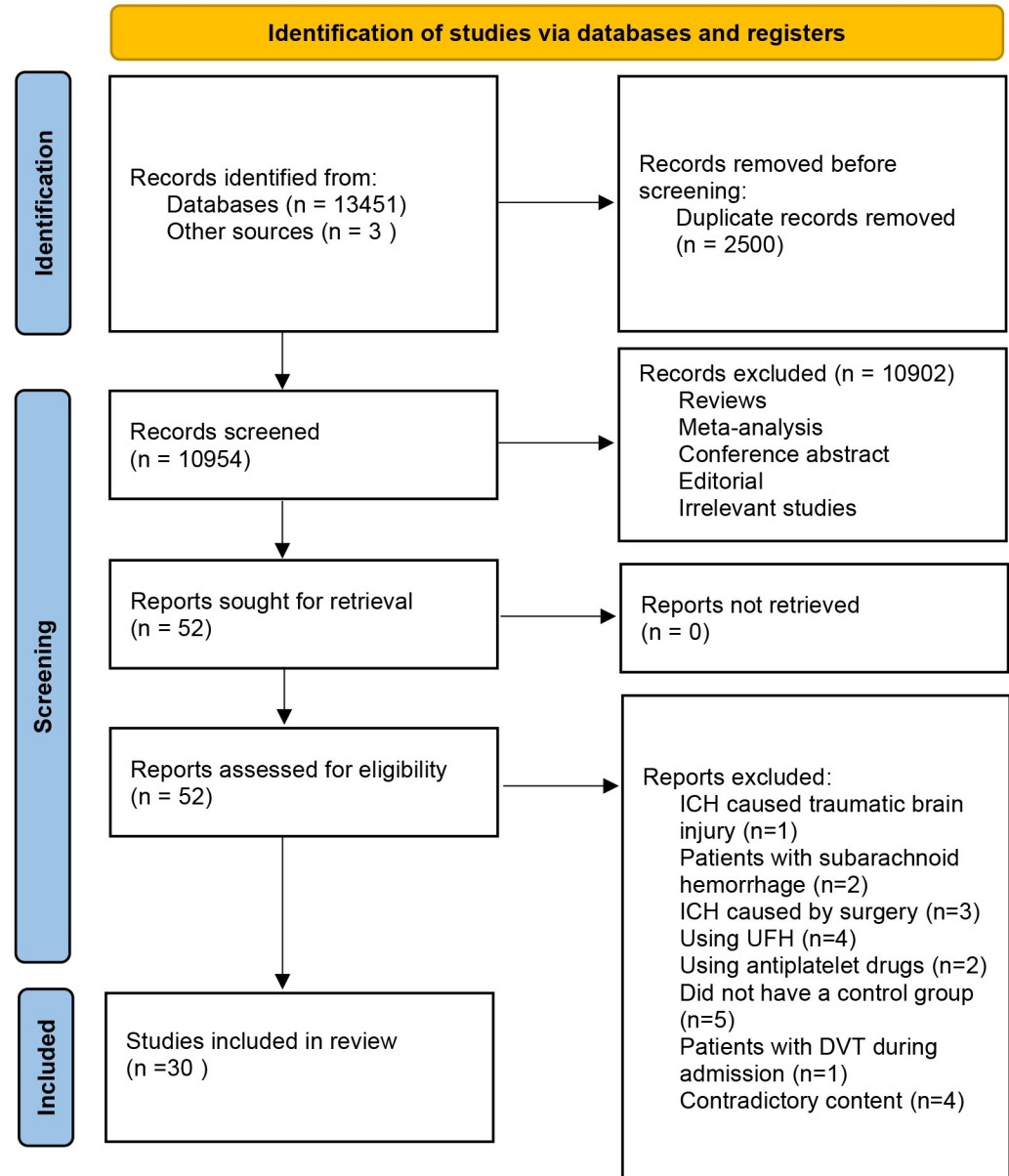

**Fig 1. Flowchart showing the study inclusion and exclusion procedure.**

**Table 1. Features of enrolled articles.**

| Study | Design | Country | Type of ICH | Participants | Operation | Treatment | Control | Dosage | Time of onset of treatment | Duration of treatment | DVT diagnosis | PE diagnosis | Hematoma progression diagnosis | Follow-up duration |
|---|---|---|---|---|---|---|---|---|---|---|---|---|---|---|
| Zhao 2020 | RCT | China | Unspecified ICH | 70 | Yes | LMWH | IPC+GCS | 4000IU QD | Post operation 3rd day | ≤10 days | Clinical | NR | NR | 14 days |
| Ni 2018 | Cohort study | China | Spontaneous ICH | 104 | No | LMWH (enoxaparin)+IPC | IPC | 0.4ml QD | Admission 4th day | 7 days | NE | NE | CT | 7 days |
| Yu 2015 | RCT | China | Unspecified ICH | 103 | No | LMWH | No LMWH | 4000IU QD | NR | 14 days | Doppler | CT | NR | 14 days |
| Yin 2019 | RCT | China | Unspecified ICH | 86 | No | LMWH+IPC+GCS | IPC+GCS | 4000IU QD | Admission 3rd day | 5 days | Doppler | NR | CT | 14 days |
| Tang 2015 | RCT | China | Hypertensive ICH | 40 | No | LMWH (nadroparin) | No LMWH | 5000IU QD | Post-ICH 3rd/4th day | 15 days | Doppler | CTA | CT | 15 days |
| Mo 2021 | RCT | China | Unspecified ICH | 80 | No | LMWH+IPC | IPC | 57IU/kg QD | | 14 days | NR | NR | NR | 14 days |
| Xu 2019 | RCT | China | Unspecified ICH | 30 | No | LMWH | No LMWH | 0.4ml | NR | NR | Clinical | NE | NE | NR |
| Qian 2012 | RCT | China | Hypertensive ICH | 60 | No | LMWH (nadroparin)+GCS | No LMWH | 0.4ml QD | NR | 14 days | Doppler | NR | CT | 14 days |
| Feng 2021 | RCT | China | Unspecified ICH | 150 | Yes | LMWH (nadroparin)+IPC | IPC | 4100IU QD | Post operation 3rd day | 10 days | Doppler | NR | NR | 10 days |
| Li 2011 | RCT | China | Hypertensive ICH | 60 | Yes | LMWH (nadroparin)+GCS | GCS | 1025IU/10kg Q12H | Post operation 3rd day | 10 days | Doppler | NE | CT | 10 days |
| Jiang 2014 | RCT | China | Hypertensive ICH | 82 | Yes | LMWH+IPC | IPC | 4100IU QD | Post operation 3rd day | 14 days | Doppler/MRI/Venography | NR | NR | 21 days |
| Liu 2008 | RCT | China | Hypertensive ICH | 60 | Yes | LMWH (nadroparin)+GCS | No LMWH | 3500IU QD | NR | 14 days | Doppler | NR | NR | 14 days |
| Xia 2018 | RCT | China | Hypertensive ICH | 102 | Yes | LMWH | No LMWH | 9500/4750IU QD | NR | NR | Clinical | NR | NR | NR |
| Yang 2018 | RCT | China | Hypertensive ICH | 100 | Yes | LMWH+IPC | No LMWH | 0.2ml QD | Post operation 3rd day | ≤14 days | Doppler | NE | NE | 14 days |
| Wang 2015 | RCT | China | Hypertensive ICH | 106 | Yes | LMWH+IPC+GCS | IPC+GCS | 4100IU QD | Post operation 3rd day | Until TEG normal | Doppler | NE | NE | 3 months |
| Yang 2010 | RCT | China | Hypertensive ICH | 99 | Yes | LMWH | No LMWH | NR | Post operation 4th day | 14 days | Doppler | NR | CT | 14 days |

(Continued)

**Table 1.** (Continued)

| Study | Design | Country | Type of ICH | Participants | Operation | Treatment | Control | Dosage | Time of onset of treatment | Duration of treatment | DVT diagnosis | PE diagnosis | Hematoma progression diagnosis | Follow-up duration |
|---|---|---|---|---|---|---|---|---|---|---|---|---|---|---|
| Qin 2018 | RCT | China | Hypertensive ICH | 98 | Yes | LMWH (nadroparin)+IPC | IPC | 57IU/kg QD | Post operation 3rd day | 14 days | Doppler/MRI/Venography | NR | CT | 21 days |
| Guan 2019 | RCT | China | Hypertensive ICH | 92 | Yes | LMWH (nadroparin)+IPC | IPC | 0.4ml QD | Post operation 3rd day | 14 days | Doppler | CTA | CT | 6 months |
| Zhang 2017 | Cohort study | China | Unspecified ICH | 150 | No | LMWH+IPC+GCS | IPC+GCS | 4000IU QD | Admission 3rd day | NR | Doppler | NE | CT | 14 days |
| Chen 2019 | Cohort study | China | Unspecified ICH | 120 | No | LMWH | No LMWH | 0.4ml | Admission 3rd day | 14 days | NR | NR | NR | 14 days |
| Gu 2014 | Cohort study | China | Spontaneous ICH | 94 | No | LMWH (nadroparin) | IPC | 0.4ml QD | Admission 4th day | 10 days | Doppler | NE | CT | 14 days |
| Wu 2022 | Cohort study | China | Spontaneous ICH | 91 | Yes | LMWH (enoxaparin)+IPC | IPC | 4000IU QD | Post operation 4th day | 7–14 days | Doppler | NR | CT | 28 days |
| Sun 2017 | Cohort study | China | Hypertensive ICH | 100 | Yes | LMWH+IPC | IPC | 4100IU QD | Post operation 3rd day | 4 days | Clinical | NE | NE | 4 days |
| Lu 2021 | Cohort study | China | Unspecified ICH | 82 | Yes | LMWH+IPC | IPC | 4100IU QD | Post operation 3rd day | 14 days | NR | NE | NE | 21 days |
| Yu 2022 | RCT | China | Unspecified ICH | 42 | Yes | LMWH | No LMWH | 3000IU QD | Admission 4th day | 7d days | NR | NE | NE | 7 days |
| Sui 2022 | RCT | China | Unspecified ICH | 50 | Yes | LMWH+IPC | IPC | 4100IU QD | Post operation 0 day | 14d days | Clinical | NE | NE | 14 days |
| Paciaroni 2020 | RCT | Italy | Spontaneous ICH | 73 | No | LMWH (enoxaparin) | No LMWH | 0.4 ml | NR | (10±1) days | Doppler | CTA | CT | 90 days |
| Song 2021 | Cohort study | China | Hypertensive ICH | 98 | No | LMWH+IPC+GCS | IPC+GCS | 4000IU QD | NR | 5 days | NR | NR | NR | 5 days |
| Tetri 2008 | Cohort study | Finland | Spontaneous ICH | 407 | No | LMWH (enoxaparin) | No LMWH | 20mg QD | Post-ICH 2nd day | 8 (5–12) days | NR | NR | CT | 3 months |
| Orken 2009 | RCT | Türkiye | Spontaneous ICH | 75 | No | LMWH (enoxaparin) | GCS | 40mg QD | Admission 3rd day | NR | Doppler | CTA | CT | 21 days |

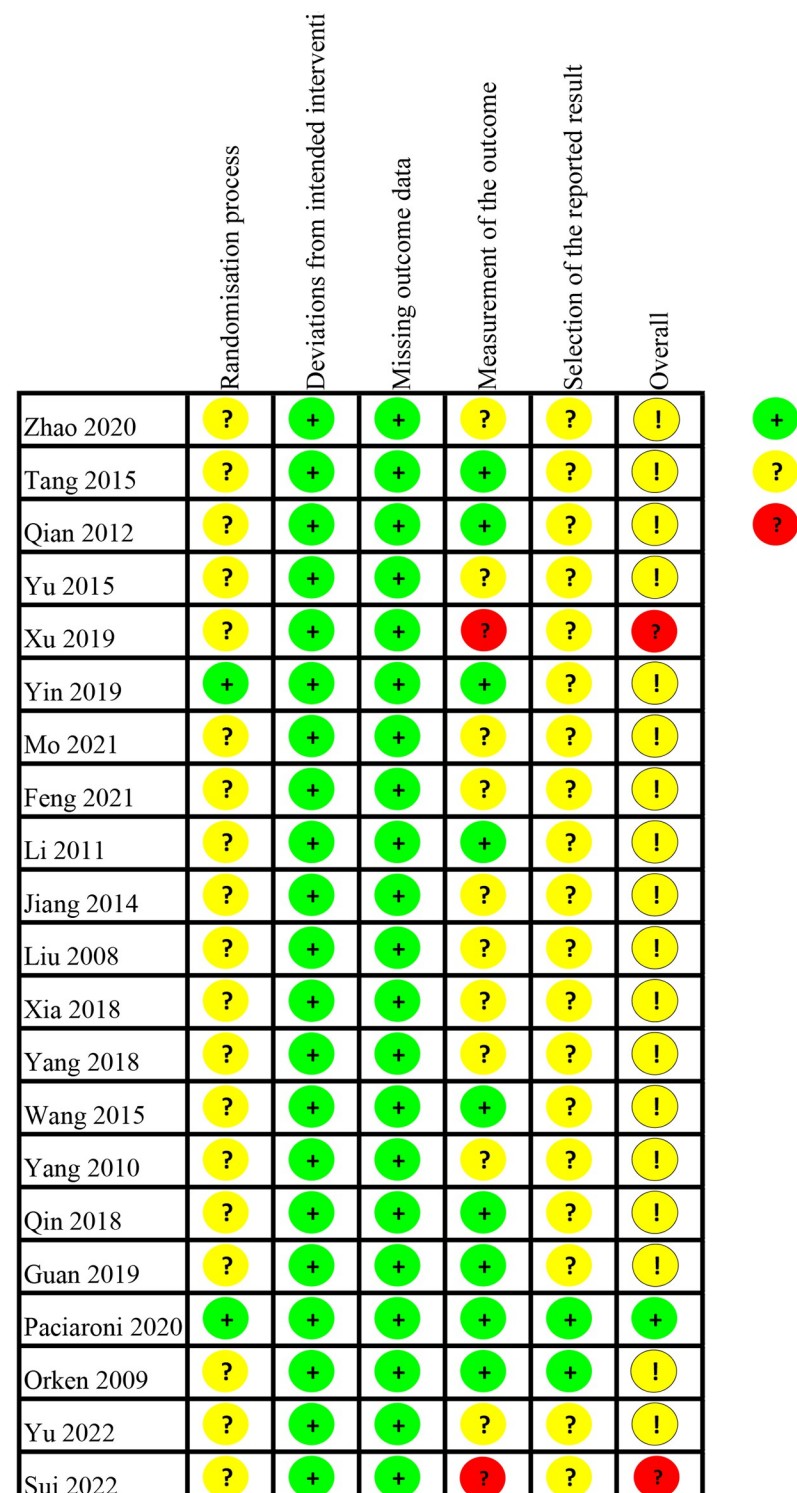

**Fig 2. Quality assessment of randomized controlled trials.**

**Table 2. Quality evaluation of non-randomized controlled trials.**

| Study | Selection | | | | Comparability | Outcome | | | |
|---|---|---|---|---|---|---|---|---|---|
| | Representativeness of the exposed cohort | Selection of the non exposed cohort | Ascertainment of exposure | Demonstration that outcome of interest was not present at start of study | Comparability of cohorts on the basis of the design or analysis | Assessment of outcome | Was follow-up long enough for outcomes to occur | Adequacy of follow up of cohorts | Total scores |
| Chen 2019 | ★ | ★ | ★ | ★ | ★★ | ☆ | ★ | ☆ | 7 |
| Zhang 2017 | ★ | ★ | ★ | ★ | ★★ | ★ | ★ | ☆ | 8 |
| Gu 2014 | ★ | ★ | ★ | ★ | ★★ | ★ | ★ | ☆ | 8 |
| Wu 2022 | ★ | ★ | ★ | ★ | ★★ | ★ | ★ | ☆ | 8 |
| Sun 2017 | ★ | ★ | ★ | ★ | ★★ | ☆ | ★ | ☆ | 7 |
| Lu 2021 | ★ | ★ | ★ | ★ | ★★ | ☆ | ★ | ☆ | 7 |
| Tetri 2008 | ★ | ★ | ★ | ★ | ☆☆ | ★ | ★ | ☆ | 6 |
| Song 2021 | ★ | ★ | ★ | ★ | ★★ | ☆ | ★ | ☆ | 7 |
| Ni 2018 | ★ | ★ | ★ | ★ | ★★ | ★ | ★ | ☆ | 8 |

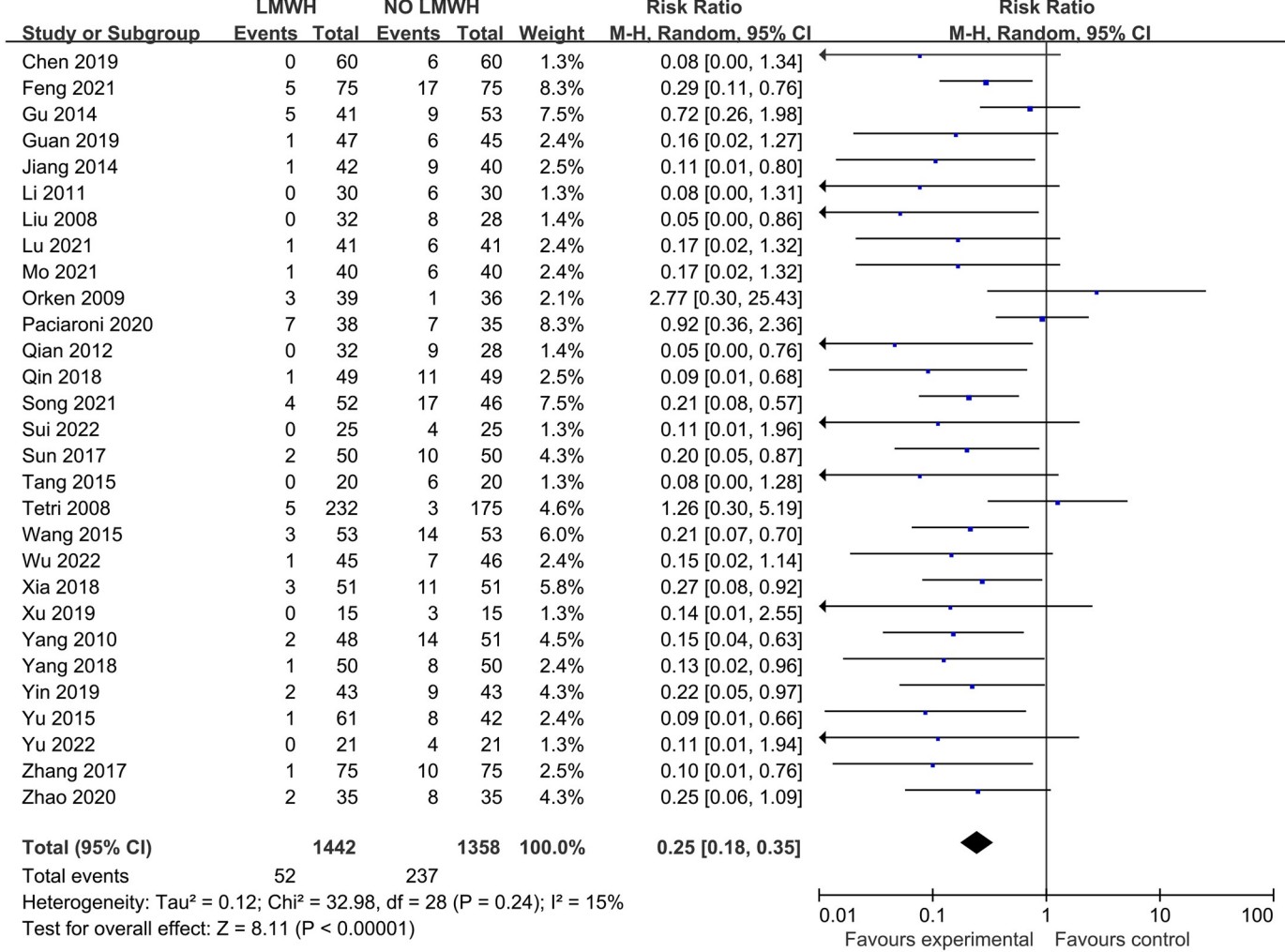

**Fig 3. Effect of LMWH on DVT.**

articles were excluded because of the following reasons: patients with traumatic ICH (one study) [8]; patients with subarachnoid hemorrhage (two studies) [9, 10]; patients with ICH due to surgery (three studies) [11–13]; use of UFH (four studies) [14–17]; use of antiplatelet drugs (two studies) [18, 19]; lack of control group (five studies) [20–24]; patients with DVT during admission (one study) [25]; contradictory content (four studies) [26–29].

### 3.2 Study quality

Fig 2 and Table 2 outline quality data for those 30 included articles. Among the 21 RCTs, two were rated as high risk of overall bias since they used clinical diagnosis to measure the outcome. Also, most articles showed certain concerns in overall bias. Among the nine non-RCTs, eight were adjudged to be high quality, only one was of moderate quality.

### 3.3 Study outcomes

Altogether 30 articles (RCTs and non-RCTs) and 2904 patients were analyzed. Amongst these studies, 13 involved hypertensive ICH patients, 6 included spontaneous ICH cases, while 11 involved unspecified ICH patients. Also, 16 studies enrolled patients after ICH operation. Thromboprophylaxis was initiated on the 3rd to 4th day after hospitalization or ICH operation and continued for 14 days in most studies. In the majority of included studies, the dosing

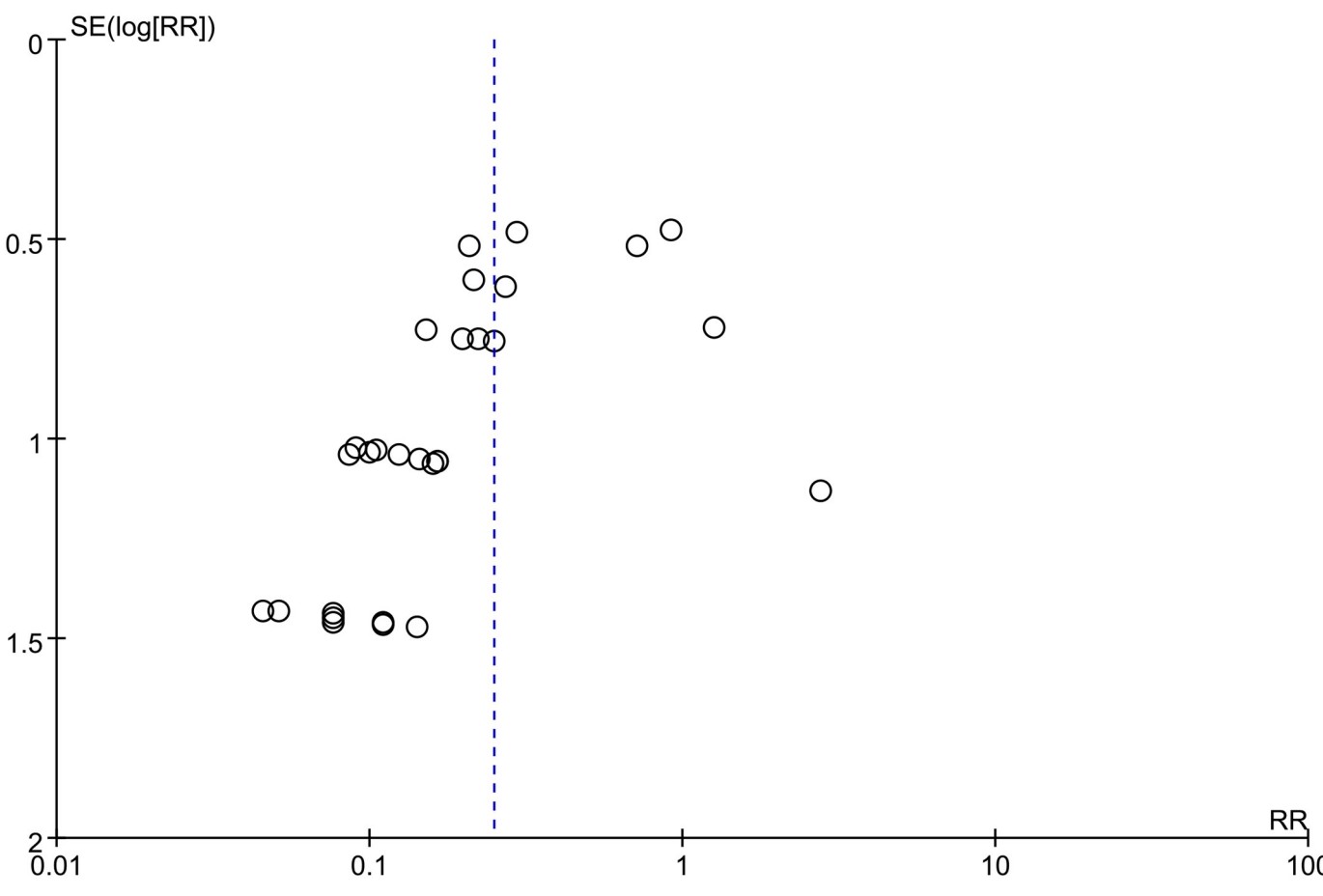

**Fig 4. Funnel plot illustrating the role of LMWH in DVT.**

regimen of LMWH was generally 0.4 ml daily (enoxaparin 4000 IU per day, nadroparin 4100 IU per day).

There was no obvious heterogeneity for the effect of LMWH on DVT among 29 studies involving 2800 patients ($I^2$ = 15%; p-value = 0.24). In comparison with control group, LMWH group showed the significantly reduced DVT (3.7% vs. 17.5%; RR, 0.25; 95% CI, 0.18–0.35; p-value<0.00001) (Fig 3). Meanwhile, asymmetry could be seen from funnel plot of DVT (Egger's test, p-value = 0.001) (Fig 4). The Filled meta-analysis performed by trim-and-fill method involved 29 articles, which conformed to initial analysis. S1–S3 Figs summaries the subgroup analysis results.

The effect of LMWH on PE, investigated based on 17 studies that involved 1807 patients, suggested the absence of obvious heterogeneity across those enrolled articles ($I^2$ = 0%; p-value = 1.00). LMWH group demonstrated significantly reduced PE in comparison with control group (0.4% vs. 3.2%; RR, 0.29; 95% CI, 0.14–0.57; p-value = 0.0003) (Fig 5). The funnel plot analysis of PE showed an asymmetrical shape (Egger's test, p-value = 0.001) (Fig 6). The Filled meta-analysis performed by trim-and-fill method involved 17 articles, which conformed to initial analysis. S4–S6 Figs summarizes the subgroup analyses.

The effect of LMWH on hematoma progression was evaluated from 18 studies with 1819 patients and obvious heterogeneity was not found across diverse studies ($I^2$ = 0%; p-value = 0.84). Further, in comparison with control group, LMWH group exhibited the non-significant increase in hematoma progression (3.8% vs. 3.4%; RR, 1.06; 95% CI, 0.68–1.68; p-value = 0.79) (Fig 7). Asymmetry was observed from funnel plot of hematoma progression (Egger's test, p-value = 0.03) (Fig 8). The Filled meta-analysis performed by trim-and-fill method involved 18 articles, which conformed to initial analysis. S7–S9 Figs displays the subgroup analyses.

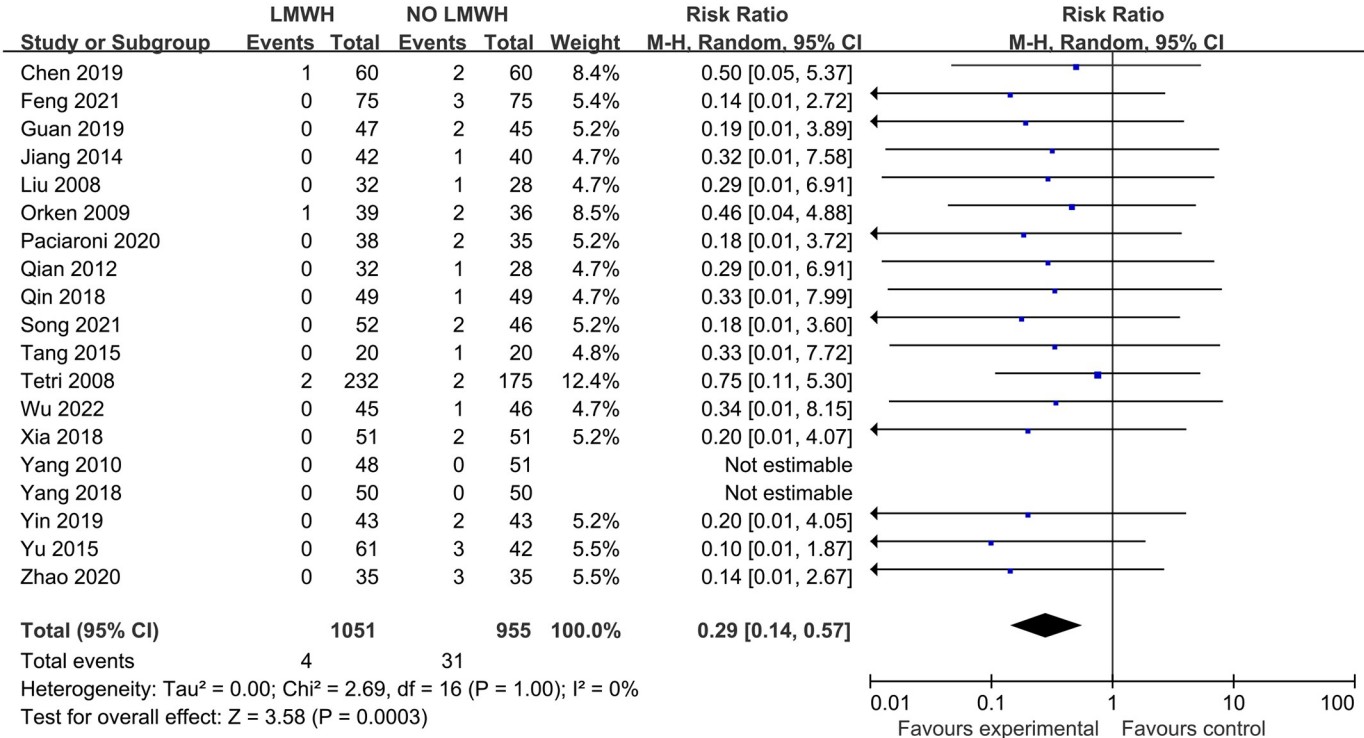

**Fig 5. Effect of LMWH on PE.**

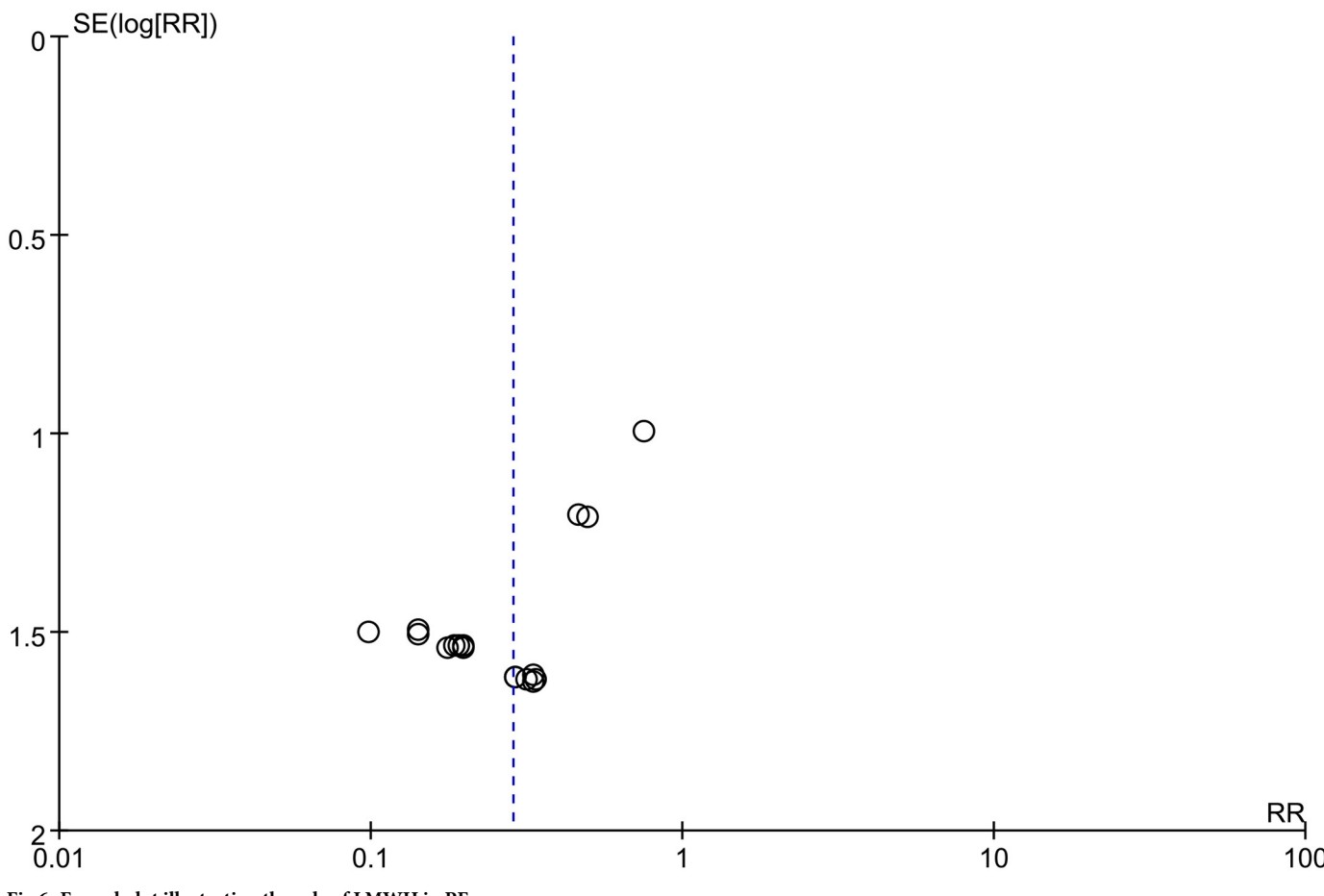

**Fig 6. Funnel plot illustrating the role of LMWH in PE.**

The meta-analysis of six studies with 530 patients on the effect of LMWH on gastrointestinal bleeding indicated the absence of obvious heterogeneity across our enrolled articles ($I^2$ = 0%; p-value = 0.60). The LMWH group showed a non-significant rise in gastrointestinal bleeding (3.6% vs. 6.1%; RR, 0.63; 95% CI, 0.31–1.28; p-value = 0.20) in comparison with control group (Fig 9). Subgroup analyses were summarized in S10–S12 Figs.

This work summarized the effect of LMWH on mortality in four articles involving 674 patients, and obvious heterogeneity was not observed across these articles ($I^2$ = 0%; p-value = 0.66). Further, mortality did not show any significant difference in LMWH versus control groups (14.1% vs. 15.8%; RR, 0.90; 95% CI, 0.63–1.28; p-value = 0.55) (Fig 10). The subgroup analyses were summarized in S13–S15 Figs.

## 4. Discussion

The clinical management of ICH patients, who develop DVT, is inconsistent due to the use of anticoagulant agents that may cause hematoma progression among the patients. Therefore, effective prophylaxis of VTE is needed in ICH patients. Generally, most of the guidelines provide weak recommendations with low quality evidence about using low-dose LMWH for ICH patients to prevent VTE [2–5]. Our findings demonstrate that in ICH patients, LMWH prophylaxis for VTE is related to the markedly reduced DVT and PE and the non-significantly

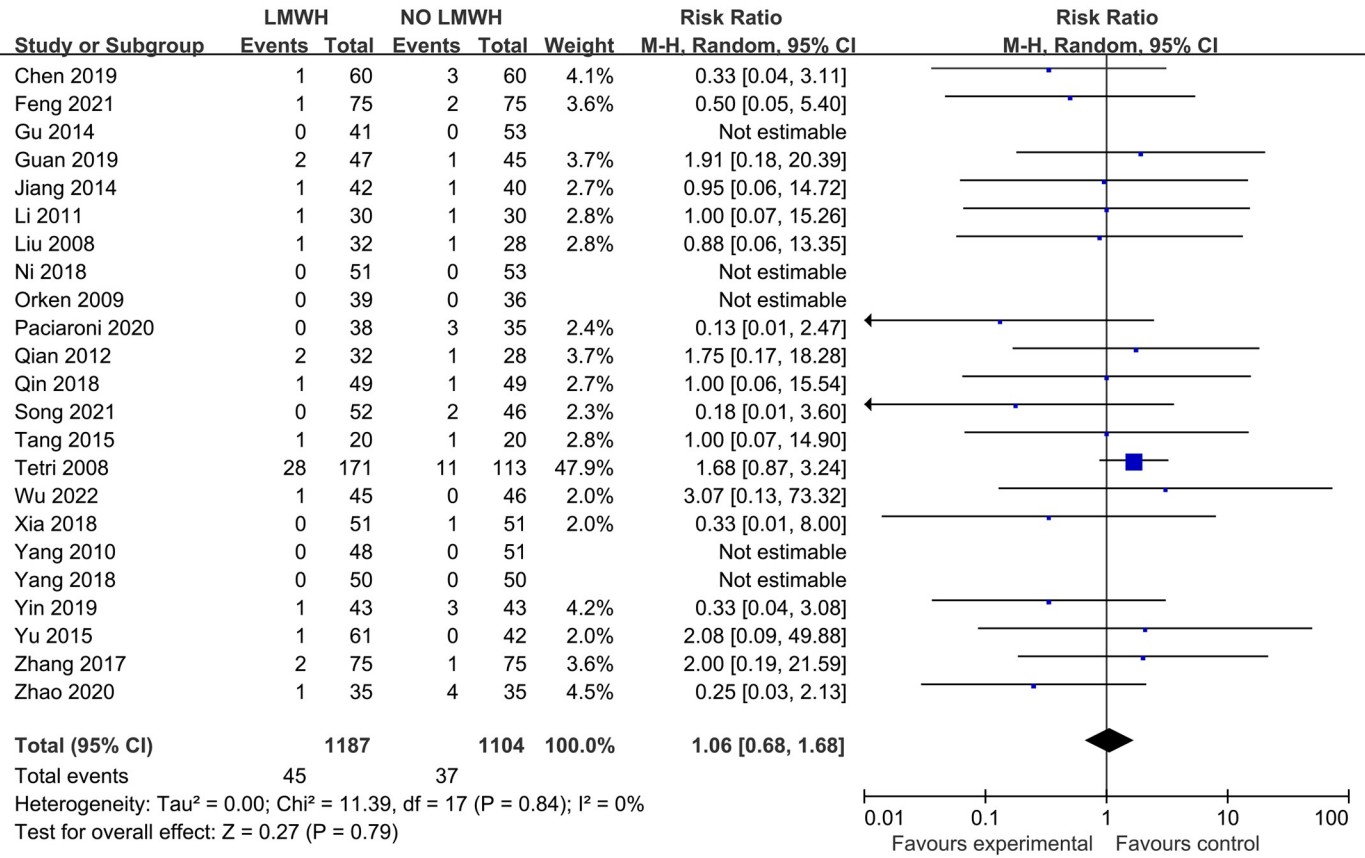

**Fig 7. Effect of LMWH on hematoma progression.**

increased hematoma progression, and gastrointestinal bleeding. One of the limitations of anti-coagulant agents in ICH patients is the increased risk of mortality. However, our data showed that mortality did not exhibit any significant difference between the LMWH and control groups.

According to the results of our meta-analysis, LMWH was initiated on the 3rd to 4th day after admission or operation and continued for 14 days in most studies. The dosing regimen of LMWH was generally 0.4 ml daily (enoxaparin 4000 IU per day, nadroparin 4100 IU per day). Consequently, early administration of low-dose LMWH is both effective and safe in ICH patients. This information may assist doctors in making clinical decisions. To further support these findings, we analyzed three subgroups, which strengthened our findings. Because we believed that whether ICH patients accept surgical treatment may affect the outcome, so we performed a subgroup analysis. In a previous meta-analysis on 4 articles, applying heparin for preventing VTE is related to the markedly reduced PE [30]. The present data is in good agreement with a recent meta-analysis that includes 28 studies and 3697 patients and demonstrates that heparin is effective and safe among ICH patients [31]. A study with 68 ICH patients suggests that heparin initiation on day two is correlated with the more markedly reduced PE than on day four or day ten [14]. However, more investigations are warranted for determining the best way to prevent VTE among ICH patients.

Although our meta-analysis offers valuable information on LMWH in preventing venous thromboembolism among ICH patients, there are certain limitations, e.g., small sample size,

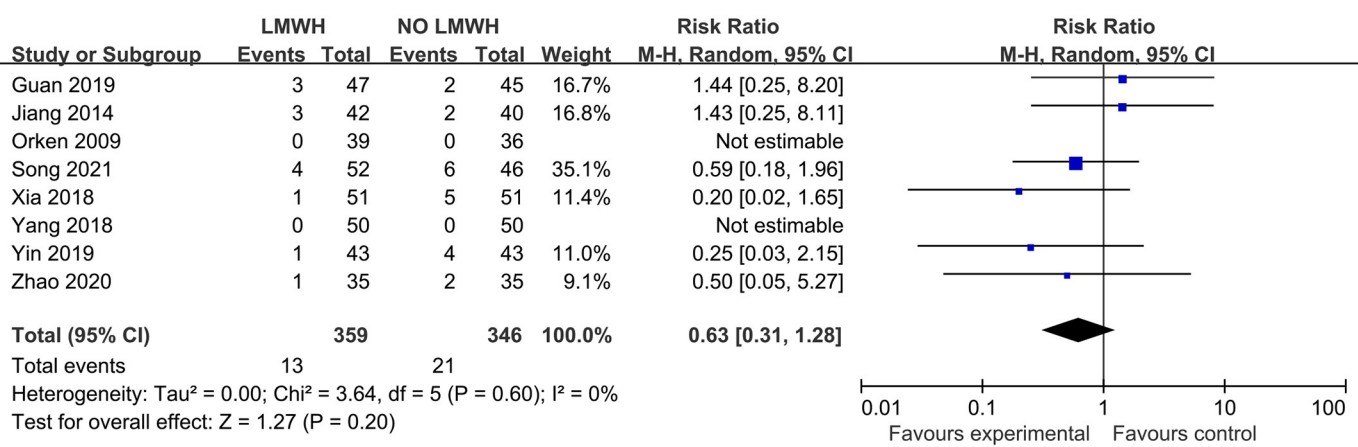

**Fig 8. Funnel plot illustrating the role of LMWH in hematoma progression.**

| Study or Subgroup | LMWH Events | Total | NO LMWH Events | Total | Weight | Risk Ratio M-H, Random, 95% CI |
|---|---|---|---|---|---|---|
| Guan 2019 | 3 | 47 | 2 | 45 | 16.7% | 1.44 [0.25, 8.20] |
| Jiang 2014 | 3 | 42 | 2 | 40 | 16.8% | 1.43 [0.25, 8.11] |
| Orken 2009 | 0 | 39 | 0 | 36 | | Not estimable |
| Song 2021 | 4 | 52 | 6 | 46 | 35.1% | 0.59 [0.18, 1.96] |
| Xia 2018 | 1 | 51 | 5 | 51 | 11.4% | 0.20 [0.02, 1.65] |
| Yang 2018 | 0 | 50 | 0 | 50 | | Not estimable |
| Yin 2019 | 1 | 43 | 4 | 43 | 11.0% | 0.25 [0.03, 2.15] |
| Zhao 2020 | 1 | 35 | 2 | 35 | 9.1% | 0.50 [0.05, 5.27] |
| | | | | | | |
| **Total (95% CI)** | | 359 | | 346 | 100.0% | **0.63 [0.31, 1.28]** |
| Total events | 13 | | 21 | | | |

Heterogeneity: Tau² = 0.00; Chi² = 3.64, df = 5 (P = 0.60); I² = 0%
Test for overall effect: Z = 1.27 (P = 0.20)

**Fig 9. Effect of LMWH on gastrointestinal bleeding.**

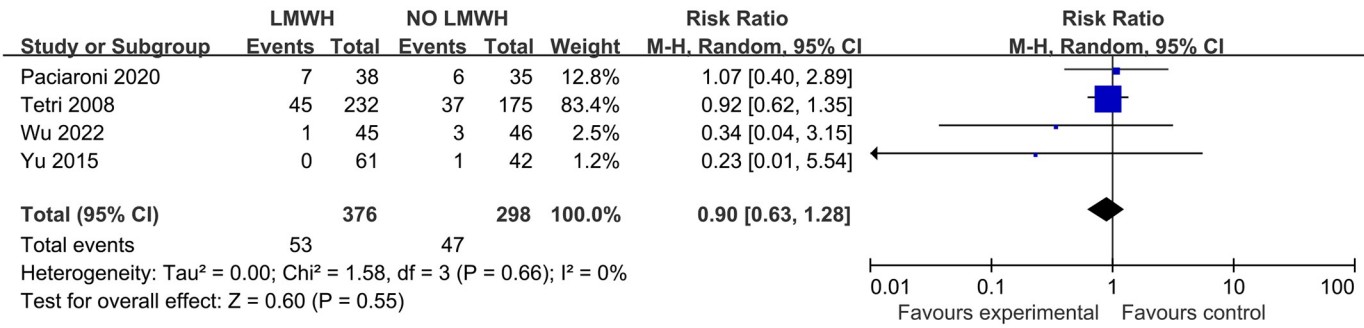

**Fig 10. Effect of LMWH on mortality.**

inclusion of non-randomized studies, concerns about overall bias, relatively short follow-up duration, and lack of long-term outcomes. Hence, additional large, multicenter, high-quality RCTs are necessary to validate the findings and inform clinical practice.

## Supporting information

**S1 Checklist. PRISMA 2020 checklist.**
(DOCX)

**S1 File. ALL studies identified in the literature search.**
(XLSX)

**S1 Table. Search details.**
(DOCX)

**S2 Table. List of articles included in the meta-analysis.**
(DOCX)

**S3 Table. Summary of outcomes in enrolled study.**
(DOCX)

**S4 Table. Quality assessment of randomized controlled trials.**
(DOCX)

**S1 Fig. Subgroup analysis stratified by study design: Effect of LMWH on DVT.**
(TIF)

**S2 Fig. Subgroup analysis stratified by ICH type: Effect of LMWH on DVT.**
(TIF)

**S3 Fig. Subgroup analysis stratified by ICH treatment type: Effect of LMWH on DVT.**
(TIF)

**S4 Fig. Subgroup analysis stratified by study design: Effect of LMWH on PE.**
(TIF)

**S5 Fig. Subgroup analysis stratified by ICH type: Effect of LMWH on PE.**
(TIF)

**S6 Fig. Subgroup analysis stratified by ICH treatment type: Effect of LMWH on PE.**
(TIF)

**S7 Fig. Subgroup analysis stratified by study design: Role of LMWH in hematoma progression.**
(TIF)

**S8 Fig. Subgroup analysis stratified by ICH type: Effect of LMWH on hematoma progression.**
(TIF)

**S9 Fig. Subgroup analysis stratified by ICH treatment type: Effect of LMWH on hematoma progression.**
(TIF)

**S10 Fig. Subgroup analysis stratified by study design: Effect of LMWH on gastrointestinal bleeding.**
(TIF)

**S11 Fig. Subgroup analysis stratified by ICH type: Effect of LMWH on gastrointestinal bleeding.**
(TIF)

**S12 Fig. Subgroup analysis stratified by ICH treatment type: Effect of LMWH on gastrointestinal bleeding.**
(TIF)

**S13 Fig. Subgroup analysis based on study design: Effect of LMWH on mortality.**
(TIF)

**S14 Fig. Subgroup analysis based on ICH type: Effect of LMWH on mortality.**
(TIF)

**S15 Fig. Subgroup analysis based on ICH treatment type: Effect of LMWH on mortality.**
(TIF)

## Author Contributions

**Conceptualization:** Haizheng Li, Zhiguo Wu.

**Data curation:** Haizheng Li, Hongyu Zhang, Baohua Qiu, Yajun Wang.

**Formal analysis:** Haizheng Li.

**Methodology:** Haizheng Li.

**Writing – original draft:** Haizheng Li.

**Writing – review & editing:** Haizheng Li.

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
