## [Decision Letter · Decision Letter 0]

24 Jul 2024

PONE-D-24-20586Low-molecular-weight heparin in the prevention of venous thromboembolism among patients with acute intracerebral hemorrhage: A meta-analysisPLOS ONE

Dear Dr. Li,

Thank you for submitting your manuscript to PLOS ONE. After careful consideration, we feel that it has merit but does not fully meet PLOS ONE’s publication criteria as it currently stands. Therefore, we invite you to submit a revised version of the manuscript that addresses the points raised during the review process.

**ACADEMIC EDITOR: **

Thank you for submitting your work to PLOS One. Based on careful review of your work and taking into account feedback from the reviewers, I would like to invite you to address the concerns raised and submit a revised version.

Whilst the study addresses an important clinical issue concerning the prevention of venous thromboembolism (VTE) in patients with acute intracerebral hemorrhage (ICH), which is a significant cause of morbidity and mortality; there are some issues that merit consideration.  By providing a meta-analysis, the manuscript seeks to provide evidence regarding the effectiveness and safety of low-molecular-weight heparin (LMWH) in this patient population.  The analysis includes 30 studies and 2904 patients.

We look forward to receiving revised version of your work. 

Major comments:

1. While the study attempts to address heterogeneity, the inclusion of both randomized controlled trials (RCTs) and non-randomized studies could introduce variability and affect the robustness of the conclusions (Page 5, Lines 84-87).

2. The study focuses primarily on short-term outcomes, which may not fully capture the long-term safety and efficacy of LMWH in ICH patients (Page 9, Lines 188-190), thereby limiting the overall scope of this work.

3. The inclusion of non-randomized studies and concerns about overall bias in some studies need to be addressed more thoroughly (Page 5, Lines 84-87).

4. While the protocol is registered in PROSPERO, additional details about deviations from the original protocol should be clearly stated (Page 4, Lines 58-59).

5. The exclusion criteria are appropriately detailed, but the justification for excluding specific types of studies (e.g., those involving other types of intracranial hemorrhage) could be better explained (Page 4, Lines 67-73).

6.  The relatively short follow-up period in some included studies may limit the ability to observe long-term outcomes (Page 9, Lines 188-190).

7. The manuscript does not provide detailed practical recommendations for implementing LMWH treatment in clinical practice (Page 10, Lines 181-185).

8. The study does not compare the effectiveness of LMWH with other potential thromboprophylactic agents or methods, which could provide a more comprehensive clinical perspective (Page 3, Lines 52-53).

9. There are a few typographical errors that need correction, such as "meta -analysis " instead of "meta-analysis" (Page 1, Line 1).

10.  Ensure consistency in the use of abbreviations and terms throughout the manuscript, such as "RCT" vs. "non-RCT" (Page 5, Lines 84-85).

11. Some references are not up-to-date, and more recent studies could be included to support the findings (Page 10, Lines 198-307).

Minor comments:

12. Abstract:

Line 1: Correct "meta -analysis " to "meta-analysis."

Line 15: Clarify the definition of "early low-dose" in the context of LMWH treatment.

13. Introduction:

Page 1, Line 2: Rephrase "life-threatening consequences, representing a significant global health burden" for clarity.

Page 2, Line 3: Provide more context on the current limitations of existing guidelines for LMWH use in ICH patients.

14. Methods:

Page 4, Line 55: Clarify the inclusion criteria for studies and justify the exclusion of non-original studies.

Page 4, Line 67: Explain the rationale for excluding studies involving other types of intracranial hemorrhage.

15. Results:

Page 7, Line 106: Elaborate on the reasons for excluding 22 articles after full-text assessment. These should also be included in the PRISMA diagram. 

Page 8, Line 132: Provide more detail on the methods used to assess heterogeneity and the implications of the findings.

16. Discussion:

Page 9, Line 164: Discuss the clinical implications of the findings in more detail, including potential changes to clinical practice guidelines.

Page 10, Line 188: Address the limitations of the study more thoroughly, including the potential impact of short follow-up duration and inclusion of non-randomized studies.

17. Conclusion:

Page 10, Line 190: Emphasize the need for large, multicentric, and high-quality RCTs to validate the findings and inform clinical practice.

We look forward to receiving your revised manuscript.

Kind regards,

Sonu Bhaskar, MD PhD

Academic Editor

PLOS ONE

2. PLOS requires an ORCID iD for the corresponding author in Editorial Manager on papers submitted after December 6th, 2016. Please ensure that you have an ORCID iD and that it is validated in Editorial Manager. To do this, go to ‘Update my Information’ (in the upper left-hand corner of the main menu), and click on the Fetch/Validate link next to the ORCID field. This will take you to the ORCID site and allow you to create a new iD or authenticate a pre-existing iD in Editorial Manager. Please see the following video for instructions on linking an ORCID iD to your Editorial Manager account: https://www.youtube.com/watch?v=_xcclfuvtxQ.

Additional Editor Comments:

See above

Reviewers' comments:

Reviewer's Responses to Questions

**Comments to the Author**

1. Is the manuscript technically sound, and do the data support the conclusions?

Reviewer #1: Partly

Reviewer #2: Yes

2. Has the statistical analysis been performed appropriately and rigorously? 

Reviewer #1: Yes

Reviewer #2: Yes

3. Have the authors made all data underlying the findings in their manuscript fully available?

Reviewer #1: Yes

Reviewer #2: No

4. Is the manuscript presented in an intelligible fashion and written in standard English?

Reviewer #1: Yes

Reviewer #2: No

5. Review Comments to the Author

Reviewer #1: Li and colleagues conducted a LMWH-focused meta-analysis to quantify LMWH’s safety and efficacy after an acute non-traumatic ICH. They manually reviewed 109,054 abstracts and 52 full-text articles to identify and analyze 30 prospective intervention trials (randomized and non-randomized). The key finding is that LMW is a safe (no rise in hematoma expansion / rebleed) and efficacious (reduced VTE) when initiated in the first 4 days of either hospitalization or surgical intervention.

Overall the study is well-done and well-reported. I have no major concerns about the overall scientific soundness of the work.

The introduction and discussion needs to be a bit better about conveying the purpose and unique value proposition of the present study compared to past meta-analyses. For example, the authors cite some discrepancies in guidelines by ASA (LMWH/UFH within 1-4 days) and HSFC (LMWH after 2 days). But this study doesn’t help in this regard because looked at any patient who got LMWH within 4 days. Also, they cite the low general quality of evidence. But compared to prior meta-analyses looking at either LMWH or UFH (Chi et al – reference #30), how does an LMWH-focused meta-analysis raise the level of evidence?

Reviewer #2: The authors present a comprehensive systematic review and meta-analysis on a critical issue of LMWH use in patients with ICH for VTE prophylaxis. The analysis was performed using PRISMA standards, and its protocol was provisionally registered in a PROSPERO. The paper is clear and contains essential findings. It may be accepted after clarification of a few issues.

Abstract

-Please clarify the search strategy and inclusion criteria in the Methods section.

Methods section

Search strategy and screening criteria

-Please state what designs of the studies were eligible for, including (RCT and cohort).

-Please clarify the type of ICH in the inclusion criteria to distinguish it from posttraumatic and postoperational.

-Please clarify if any automatic tools were used to screen titles and abstracts and exclude duplications.

Quality evaluation

-Due to the presence of different approaches for interpretation of the NOS score, please clarify your own.

Endpoints

-Please clarify the definitions of all outcomes. What is the difference between hematoma expansion and rebleeding?

-According to the analysis of the risk of selection bias, I can suggest that GI bleeding and hematoma expansion/rebleeding were not considered primary outcomes. Please clarify and divide outcomes into primary and secondary. Please note that all primary outcomes need a separate analysis for publication bias, as you already performed for DVT, PE, and hematoma expansion/rebleeding.

Statistical analysis

-Do you mean p<0.01 for heterogeneity?

-Please clarify a threshold for I2 that represents heterogeneity.

-Please clarify whether under treatment type you mean surgery or not surgery.

-Please consider subgroup analysis for the LMWH initiation time, which is crucial for ICH.

-Also, it is interesting to look at the effect of LMWH in terms of different comparators (IPC alone, IPC + ESC, no prophylaxis).

Results

Study screening

-What do you mean by the use of heparin in the excluded studies? Unfractionated heparin? Please consider that you do not have such an exclusion criterion in the methods section.

Study outcomes

-You stated about the non-significant increase in GI bleeding; however, the forest plot favors LMWH. Please check this statement.

Discussion

-Please discuss the clinical relevance of your findings in the context of the lack of influence of LMWH on mortality.

Figure 1

Please design a flowchart in accordance with the PRISMA standard and describe what other sources were used to find 3 records (reference list of relevant papers? authors' archive?)

Supplementary

-Please check your query. Using #1 for Pubmed, I found 60280 results that significantly exceed the stated 13451.

6. PLOS authors have the option to publish the peer review history of their article (what does this mean?). If published, this will include your full peer review and any attached files.

Reviewer #1: No

Reviewer #2: **Yes: **Kirill Lobastov

---

## [Author Response · Author response to Decision Letter 0]

10 Aug 2024

Thank you for your letter and for your comments concerning our manuscript. Our respond to editor and reviewer comments are detailed in rebutter letter.

---

## [Decision Letter · Decision Letter 1]

22 Sep 2024

PONE-D-24-20586R1Low-molecular-weight heparin in the prevention of venous thromboembolism among patients with acute intracerebral hemorrhage: A meta-analysisPLOS ONE

Dear Dr. Li,

Thank you for submitting your manuscript to PLOS ONE. After careful consideration, we feel that it has merit but does not fully meet PLOS ONE’s publication criteria as it currently stands. Therefore, we invite you to submit a revised version of the manuscript that addresses the points raised during the review process.

We look forward to receiving your revised manuscript.

Kind regards,

Sonu Bhaskar, MD PhD

Academic Editor

PLOS ONE

Journal Requirements:

Additional Editor Comments:

Thank you for submitting the revised version of your manuscript. While there have been improvements, there are still a few issues that need to be addressed before the manuscript can be considered further. I kindly invite you to respond to the reviewers' remaining queries and submit a revised version of your manuscript for further evaluation.

Reviewers' comments:

Reviewer's Responses to Questions

**Comments to the Author**

1. If the authors have adequately addressed your comments raised in a previous round of review and you feel that this manuscript is now acceptable for publication, you may indicate that here to bypass the “Comments to the Author” section, enter your conflict of interest statement in the “Confidential to Editor” section, and submit your "Accept" recommendation.

Reviewer #1: (No Response)

Reviewer #3: (No Response)

Reviewer #4: (No Response)

2. Is the manuscript technically sound, and do the data support the conclusions?

Reviewer #1: Partly

Reviewer #3: Yes

Reviewer #4: Yes

3. Has the statistical analysis been performed appropriately and rigorously? 

Reviewer #1: Yes

Reviewer #3: Yes

Reviewer #4: Yes

4. Have the authors made all data underlying the findings in their manuscript fully available?

Reviewer #1: Yes

Reviewer #3: Yes

Reviewer #4: No

5. Is the manuscript presented in an intelligible fashion and written in standard English?

Reviewer #1: Yes

Reviewer #3: Yes

Reviewer #4: No

6. Review Comments to the Author

Reviewer #1: (No Response)

Reviewer #3: The Authors present the results of a meta-analysis related to a frequently encountered clinical problem: is it safe to use prophylactic LMWH for thrombosis prevention in spontaneous ICH? And how long after ICH should LMWH be started?

Comments to the Authors:

1. The meta-analysis inclusion criteria only states: studies reporting patients with ICH. It should be corrected to spontaneous ICH as other ICH types are listed in the exclusion criteria.

2. Hematoma expansion and rebleeding are basically the same. I would recommend using the term “hematoma progression” instead. Also correct in Table 1.

3. In the Results section: four studies were excluded due to heparin use. I’m guessing you mean unfractionated heparin. Also correct in Figure 1.

4. Please elaborate why did you include 11 studies with unspecified ICH if your primary focus was on spontaneous ICH.

5. The risk of GI bleeding was non-significantly lower in the LMWH group based on the reported numbers and Forest plot. Please correct in the Results section.

Reviewer #4: The authors conducted a meta-analysis to evaluate the effectiveness of low molecular weight heparin (LMWH) in on preventing venous thromboembolism (VTE) among ICH patients. While the authors should be commended for conducting a thorough analysis, it is my humble opinion that this manuscript will benefit immensely from copyediting to address the numerous grammatical errors and improve overall clarity.

The search strategy (the actual terms used to conduct the search) utilized by the authors is not clearly communicated. Moreover, the search strategy seems to miss ICH, which is the primary patient population for which this research was conducted. The authors only mentioned that ICH was added later in the manuscript. Given that a systematic review should be reproducible, it is advised that the authors employ the service of a librarian to improve their systematic search and make it more reproducible.

Given that the studies evaluating most of the adverse outcomes are few, it will be advisable to evaluate a composite outcome of any major bleeding.

7. PLOS authors have the option to publish the peer review history of their article (what does this mean?). If published, this will include your full peer review and any attached files.

Reviewer #1: No

Reviewer #3: No

Reviewer #4: No

---

## [Author Response · Author response to Decision Letter 1]

24 Sep 2024

Thank you for your letter and for your comments. Those comments are all valuable and very helpful for revising and improving our paper, as well as the important guiding significance to our researches. We have studied comments carefully and have made correction which we hope meet with approval. We tried our best to improve the manuscript and made some changes in the manuscript. We appreciate for your warm work earnestly, and hope that the correction will meet with approval. Once again, thank you very much for your comments and suggestions. I appreciate your time and look forward to hearing from you soon.

---

## [Editor Report · Decision Letter 2]

26 Sep 2024

Low-molecular-weight heparin in the prevention of venous thromboembolism among patients with acute intracerebral hemorrhage: A meta-analysis

PONE-D-24-20586R2

Dear Dr. Li,

We’re pleased to inform you that your manuscript has been judged scientifically suitable for publication and will be formally accepted for publication once it meets all outstanding technical requirements.

Kind regards,

Sonu Bhaskar, MD PhD

Academic Editor

PLOS ONE

Additional Editor Comments (optional):

Thank you for submitting the revised version of your manuscript. After careful review, I am pleased to inform you that it has been accepted in its current form.
---

## [Editor Report · Acceptance letter]

7 Oct 2024

PONE-D-24-20586R2 

PLOS ONE

Dear Dr. Li, 

I'm pleased to inform you that your manuscript has been deemed suitable for publication in PLOS ONE. Congratulations! Your manuscript is now being handed over to our production team.

Kind regards, 

on behalf of

Dr. Sonu Bhaskar 

Academic Editor

PLOS ONE